

# Observed change and the extent of coherence in the Gulf Stream system

Helene Asbjørnsen[1,2], Tor Eldevik[1,2], Johanne Skrefsrud[1,2], Helen L. Johnson[3], and
Alejandra Sanchez-Franks[4]

[1]Geophysical Institute, University of Bergen, Bergen, Norway
[2]Bjerknes Centre for Climate Research, Bergen, Norway
[3]Department of Earth Sciences, University of Oxford, Oxford, United Kingdom
[4]National Oceanography Centre, Southampton, United Kingdom

**Correspondence:** Helene Asbjørnsen (h.asbjornsen@uib.no)

**Abstract.** By transporting warm and salty water poleward, the Gulf Stream system maintains a mild climate in northwestern Europe while also facilitating the dense water formation that feeds the deep ocean. The sensitivity of North Atlantic circulation to future greenhouse gas emissions seen in climate models has prompted an increasing effort to monitor the various ocean circulation components in recent decades. Here, we synthesise available ocean transport measurements from several observational

programs in the North Atlantic and Nordic Seas, as well as an ocean state estimate (ECCOv4-r4), for an enhanced understanding of the Gulf Stream and its poleward extensions as an interconnected circulation system. We see limited coherence between the records on interannual time scales, highlighting the local oceanic response to atmospheric circulation patterns and variable recirculation time scales within the gyres. On decadal time scales, we find a weakening subtropical circulation between the mid-2000s and mid-2010s, while the inflow and circulation in the Nordic Seas remained stable. Differing decadal trends in the

subtropics, subpolar North Atlantic, and Nordic Seas warrant caution in using observational records at a single latitude to infer large-scale circulation change.

## 1 Introduction

The steady supply of warm Gulf Stream water from the subtropics to subpolar latitudes is crucial for maintaining a mild, maritime climate in northwestern Europe (Kwon et al., 2010; Palter, 2015). Projected slowdown of the North Atlantic circulation

as a response to global warming (e.g., Manabe and Stouffer, 1994; Weijer et al., 2020; Sen Gupta et al., 2021) has therefore motivated extensive observational efforts to monitor and understand variability and potential future change (Cunningham et al., 2007; Mercier et al., 2015; Lozier et al., 2017; Østerhus et al., 2019; Rhein et al., 2019). Inferences about large-scale circulation change are typically made based on observing systems measuring circulation strength at carefully selected fixed locations. However, it remains unclear to what extent, and on which time scales, the extended Gulf Stream system should be considered a

meridionally coherent circulation system. Here, we use North Atlantic and Nordic Seas ocean transport measurement records to investigate meridional coherence, interannual variability, and potential trends within the Gulf Stream system.





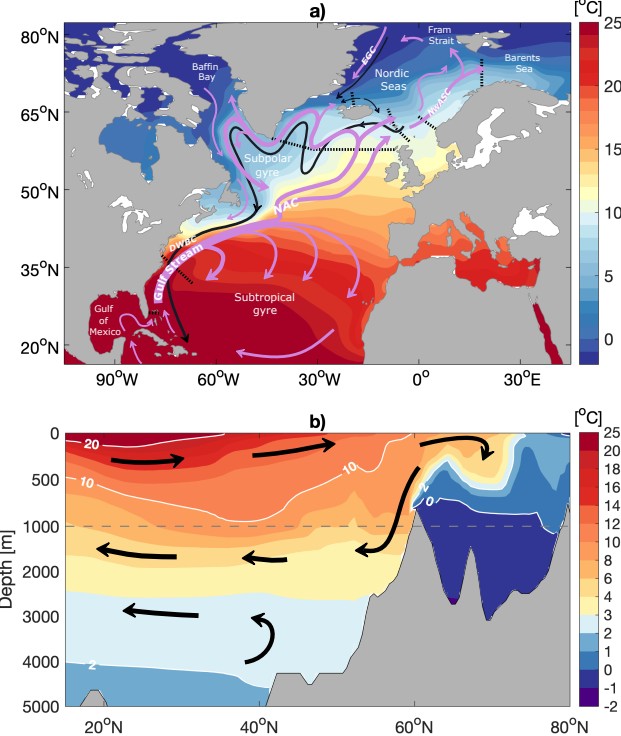

**Figure 1.** Main North Atlantic and Nordic Seas circulation features. Climatological potential temperature from the ECCOv4 ocean state estimate showing (a) the horizontal view; purple and black arrows indicate upper-ocean and deep-ocean circulation, respectively, (b) the vertical view; transect follows the WOCE A16 section until 52°N, from where it veers into the Nordic Seas over the Iceland-Scotland Ridge and toward the Fram Strait. In (a) the dashed lines show the observational sections included in the analysis. Note the nonlinear y-axis and nonlinear colorbar in (b).

As a narrow western boundary current, the Gulf Stream is a part of the subtropical gyre (Figure 1a). At approximately 35°N, the Gulf Stream separates from the coast and broadens, reaching a maximum of around 150 Sv at 60°W (Hogg, 1992). The North Atlantic Current continues as the north-eastward extension of the Gulf Stream past the Grand Banks, transporting roughly 27 Sv into the eastern subpolar North Atlantic (Roessler et al., 2015). Substantial subtropical and subpolar recirculation characterizes the North Atlantic circulation (e.g., Daniault et al., 2016). Still, almost 2/3 of the waters flowing across the Greenland-Scotland Ridge into the Nordic Seas come from the Gulf Stream (Asbjørnsen et al., 2021), highlighting the direct connection between the Gulf Stream and the Atlantic water ultimately reaching the Arctic via the Norwegian Atlantic Current. Here, we consider the North Atlantic Current and the Norwegian Atlantic Current as part of the 'Gulf Stream system', but retain the terminology 'the Gulf Stream' for the boundary current along the North American coast.

The Gulf Stream and its poleward extensions are important to the large-scale overturning circulation where water is transformed from light to dense water masses through surface heat loss and mixing at high latitudes (e.g., Mauritzen, 1996; Lozier, 2012). The Atlantic Meridional Overturning Circulation (AMOC) is quantified as the zonally integrated and vertically accumu-





lated meridional flow in the North Atlantic, of which the Gulf Stream and the extensions form the essential northward flowing
branch between 25°N and 70°N. The resulting overturning streamfunction depicts an upper-cell where warm, subtropical wa-
ter flows northward and cold North Atlantic Deep Water flows southward at depth (Figure 1b). While considerable effort has
been made to observe and understand the AMOC, it is by definition a zonally integrated view of the circulation which masks
variability in the individual currents (e.g., Lozier, 2010; Roquet and Wunsch, 2022). Both model and observational studies
show, for instance, limited coherence between subtropical and subpolar AMOC on seasonal to decadal time scales (Bingham
et al., 2007; Lozier et al., 2010; Mielke et al., 2013; Moat et al., 2020; Jackson et al., 2022). As an alternative to the integrated
AMOC view, we do a first assessment of observed variability in the northward flowing upper-ocean branches within the Gulf
Stream system.

The North Atlantic Ocean exhibits pronounced variability on a range of time scales. The dominant mode of interannual
atmospheric variability, the North Atlantic Oscillation (NAO), drives ocean circulation changes through both wind stress and
surface heat flux anomalies on interannual to decadal time scales (e.g., Eden and Willebrand, 2001; Marshall et al., 2001;
Sarafanov, 2009). The subpolar North Atlantic has distinct decadal trends in heat- and freshwater content linked to subpolar
gyre dynamics (Piecuch et al., 2017; Desbruyères et al., 2021; Fox et al., 2022). On multidecadal time scales, warm and cold
phases referred to as the Atlantic Multidecadal Variability are characterized by basin-wide sea surface temperature anomalies
with AMOC variability thought to be an important driver (Zhang et al., 2019). In addition to internal variability, externally
forced global warming is projected to slow down the AMOC over the 21st century by reducing dense water formation at
subpolar latitudes (e.g., Lique and Thomas, 2018; Weijer et al., 2020).

The observational record is relatively short considering the wide range of time scales characterizing North Atlantic vari-
ability. The AMOC strength has been measured by the RAPID (Cunningham et al., 2007) and OSNAP (Lozier et al., 2017)
cross-basin observing systems since 2004 and 2014, respectively. The Nordic Seas inflow branches and the Norwegian Atlantic
Slope Current have been monitored since the 1990s (Orvik and Skagseth, 2003b; Ingvaldsen et al., 2004; Østerhus et al., 2019).
For the interannual to decadal time scales resolved by the records so far, it remains unclear to what extent the different branches
of the Gulf Stream system will exhibit coherent variability and thus can be used to make inferences about the large-scale circu-
lation. Distinguishing naturally occurring variability from an externally forced global warming signal is furthermore a major
challenge (Baehr et al., 2008; Kelson et al., 2022).

Here, we focus on observational records of circulation strength from the Florida Current in the subtropical North Atlantic
to the Norwegian Atlantic Current in the Nordic Seas (Figure 2). We use the ECCOv4-r4 ocean state estimate to extend
the analysis back to 1992 and explore mechanisms of interannual to decadal variability. In evaluating the Gulf Stream and
its poleward extensions as an interconnected circulation system within this time period, we identify patterns of coherence
and incoherence which have implications for the interpretation of single observational records in the context of large-scale
circulation change.





**Table 1.** Ocean transport measurement records in the North Atlantic and Nordic Seas. The mean transport is the absolute value of the monthly mean volume transport. The trend is the linear trend over the respective measurement periods (negative sign denotes weakening). Records quantifying overturning strength are included at RAPID ($moc_z$), OSNAP ($moc_\sigma$), and the Greenland-Scotland Ridge (GSR OW; overflows). The notation ($moc_z$) and ($moc_\sigma$) denotes depth-space and density-space overturning strength, respectively. Significant trends are marked in bold font using the modified Mann-Kendall trend test for autocorrelated data (Hamed and Rao, 1997).

| Section | ~Latitude | Time period | Mean [Sv] | Trend [Sv/yr] | Reference |
|---|---|---|---|---|---|
| **Florida Current** | 26°N | Mar 1982 - Aug 2021 | 31.8 | **−0.0325** | Meinen et al. (2010) |
| **RAPID WBC** | 26°N | Apr 2004 - Dec. 2020 | 33.2 | −0.0357 | Smeed et al. (2018) |
| **RAPID $moc_z$** | 26°N | Apr 2004 - Mar 2020 | 16.9 | **−0.1208** | Moat et al. (2020) |
| **Oleander GS** | 36°N | Jun 1993 - Feb 2018 | 95.0 | 0.0224 | Rossby et al. (2019) |
| **OSNAP-East NAC** | 58°N | Aug 2014 - Jun 2020 | 19.2 | −0.1602 | Fu et al. (2023) |
| **OSNAP $moc_\sigma$** | 58°N | Aug 2014 - Jun 2020 | 16.4 | 0.1297 | Fu et al. (2023) |
| **GSR** | 60°N | Oct 1994 - Jul 2016 | 7.4 | **0.0125** | Østerhus et al. (2019) |
| **GSR OW** | 60°N | Jul 1997 - Apr 2017 | 5.4 | **0.0190** | Østerhus et al. (2019) |
| **Svinøy** | 62°N | Apr 1995 - May 2020 | 4.5 | −0.0024 | Orvik (2022) |
| **BSO** | 73°N | Sep 1997 - Mar 2017 | 2.0 | 0.0111 | Ingvaldsen et al. (2004) |

## 2 Methods

The strength of the ocean circulation is monitored by a number of observational arrays in the North Atlantic and Nordic Seas. In Section 2.1, we give an overview of the ocean transport measurements used in the analysis (Figure 2): the Florida Current and Western Boundary Current at 26.5°N, the Gulf Stream at the Oleander section, the North Atlantic Current at OSNAP-East, the Greenland-Scotland Ridge inflows to the Nordic Seas, the Norwegian Atlantic Current at Svinøy, and the Atlantic water inflow to the Barents Sea. While our focus is the upper-ocean circulation, we also show estimates of overturning strength (Figure 3): maximum of the overturning streamfunction at the RAPID and OSNAP sections, and overflow transports at the Greenland-Scotland Ridge. The ECCOv4-r4 ocean state estimate is described in Section 2.2, and the data treatment is explained in Section 2.3.

## 2.1 Observing systems

The Florida Current has been measured since 1982 and is the longest, near-continuous volume transport time series in the North Atlantic (Larsen and Sanford, 1985; Baringer and Larsen, 2001). The volume transport is inferred from submarine telephone cables measuring the motionally induced voltage difference across the strait between Florida and Grand Bahama Island. The 32 Sv transported by the Florida Current (Table 1) and the, on average, 4.7 Sv in the Antilles Current east of the Bahamas (Meinen et al., 2019) constitute the starting point of the Gulf Stream. Because variability in the Antilles Current is important



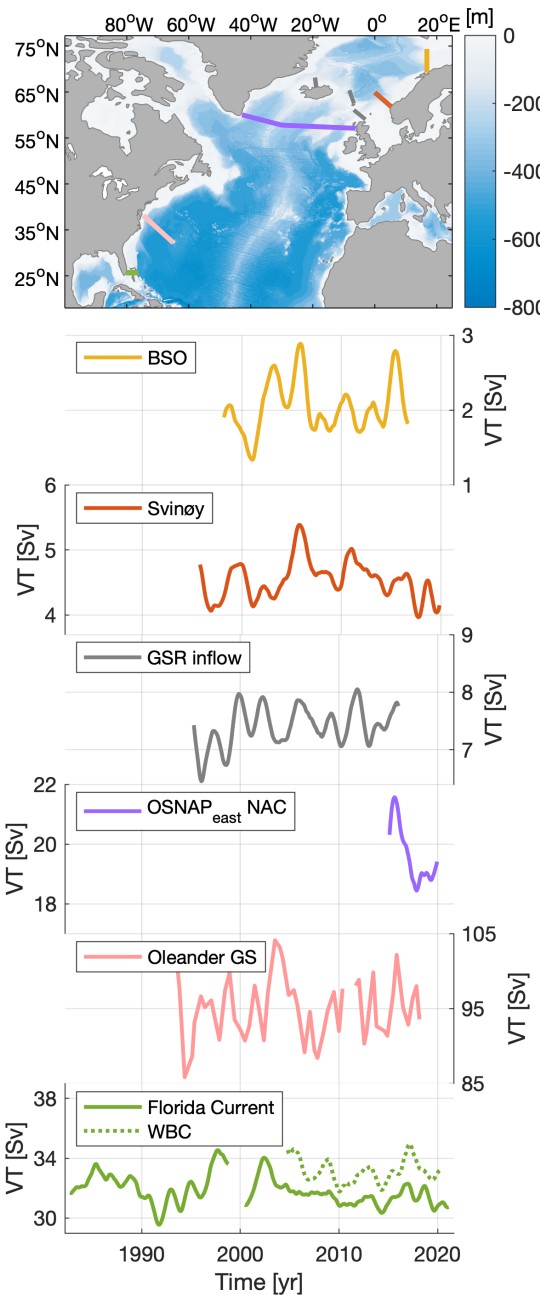

**Figure 2.** Volume transport time series for the observed Gulf Stream system. From the southernmost to the northernmost section; the Florida Current and Western Boundary Current (WBC; includes Antilles Current) at 26.5°N, the Gulf Stream (GS) component at the Oleander section, the North Atlantic Current (NAC) at OSNAP-East, the combined Faroe-Shetland Channel, Iceland-Faroe-Ridge, and Denmark Strait inflows at the Greenland-Scotland Ridge (GSR), the Norwegian Atlantic Slope Current at Svinøy, and the Barents Sea Opening (BSO) inflow. The monthly mean time series have been filtered with a 1-year low-pass triangular filter to highlight interannual variability.



for the overall variability at the western boundary (Figure 2), we include the total Western Boundary Current transport which combines the Florida Current and the Antilles Current (Smeed et al., 2018).

The RAPID-MOCHA array has been active since 2004, estimating the flow across 26.5°N (Cunningham et al., 2007; Moat et al., 2020). Considering the circulation to be in near geostrophic balance away from boundaries, the RAPID array estimates
the mid-ocean geostrophic transport from the thermal wind relation using dynamic height moorings located at the western and eastern continental shelves and on both sides of the Mid-Atlantic Ridge (McCarthy et al., 2015). The full AMOC estimate additionally relies on current meter moorings measuring the Antilles Current, cable measurements from the Florida Current, and the Ekman transport calculated from ERA5 wind stress (Hersbach et al., 2020). The AMOC strength at RAPID shown in Figure 3 is the maximum of the estimated overturning streamfunction in depth-space ($moc_z$), and thus reflects the strength of
the net upper-ocean circulation at 26.5°N.

An ADCP mounted on the container ship $CMV\ Oleander$ allows for estimating volume fluxes from velocities measured along a transect from New Jersey to Bermuda (Rossby et al., 2005). The ADCP measurements reach 250-400m depth for the 1992-2004 period, and 500-600m from 2005 and onwards (Sanchez-Franks et al., 2014). Because the measurements do not cover the full depth, the Oleander record is a volume flux for a 1 m thick layer at 52 m depth (unit; Sv/m). We here focus on
the Gulf Stream component defined as the northeastward, high-velocity core as provided in Rossby et al. (2019). Using a scale factor of 700, the total Gulf Stream transport in the 0-2000 m layer can be estimated (Rossby et al., 2014), averaging to 95 Sv (Table 1). Due to variable sampling frequency related to ship time and equipment failure, the Oleander transport is estimated in 1-year segments stepped at half-year intervals. As a result, the Oleander record has different temporal resolution than the other time series displayed in Figure 2.

The OSNAP observing system, deployed in 2014, monitors the North Atlantic circulation at subpolar latitudes (Lozier et al., 2017). The two sub-arrays OSNAP-East and OSNAP-West use densely spaced current meter and dynamic height moorings in the boundary currents and over the Reykjanes Ridge. OSNAP also relies on Argo float data, satellite altimetry, glider observations, and the surface wind field to estimate velocities and property fields away from the moorings (Li et al., 2017). Here, we use the North Atlantic Current transport across OSNAP-East (Figure 2), defined as the net transport east of 25.6°W
and above the 27.77 $kg/m^3$ isopycnal. We also show the AMOC strength in density-space ($moc_\sigma$) for the full OSNAP line (Figure 3), which quantifies water mass transformation from light to dense water north of the section.

The three inflow branches to the Nordic Seas across the Greenland-Scotland Ridge are monitored by three sub-arrays with current meter moorings at the Faroe-Shetland Channel (Berx et al., 2013), Iceland-Faroe Ridge (Hansen et al., 2015), and north of the Denmark Strait at the Hornbanki section (Jónsson and Valdimarsson, 2012). Regular CTD cruises also sample
the sections multiple times a year. For the Faroe-Shetland Channel and Iceland-Faroe Ridge, the volume transport time series combine in situ observations with satellite altimetry. On average, 2.7 Sv is transported in the Faroe-Shetland Channel, 3.8 Sv across the Iceland-Faroe Ridge, and 0.9 Sv with the Denmark Strait branch (Østerhus et al., 2019). We also show the transport of the Greenland-Scotland Ridge overflows (Figure 3; Denmark Strait and Faroe-Bank Channel overflows), which quantifies the amount of dense water formed north of the ridge and exported to the Atlantic Ocean (Hansen et al., 2016; Jochumsen et al.,
115  2017).



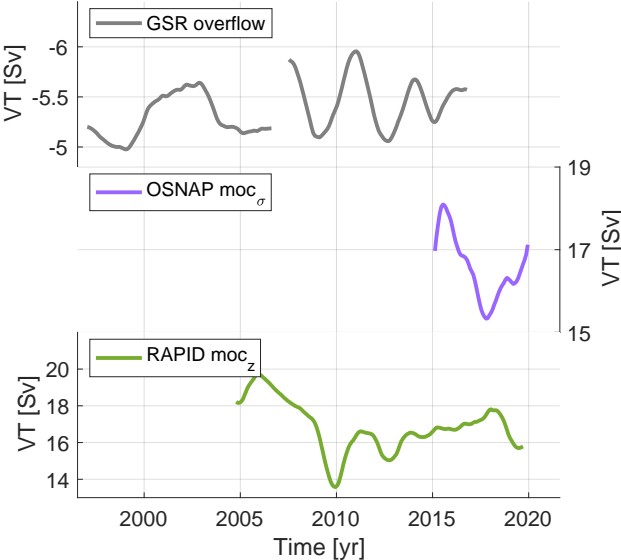

**Figure 3.** Volume transport time series quantifying overturning strength. The Denmark Strait and Faroe-Bank-Channel overflows at the Greenland-Scotland Ridge (GSR), overturning strength in density space ($moc_\sigma$) at OSNAP, and overturning strength in depth-space ($moc_z$) at RAPID. The monthly mean time series have been filtered with a 1-year low-pass triangular filter to highlight interannual variability. Note that the y-axis of the GSR overflow panel has been flipped.

North of the Greenland-Scotland Ridge, a mooring in the Norwegian Atlantic Slope Current has been measuring its variability since 1995 (Orvik and Skagseth, 2003a; Orvik, 2022). The mooring is located at the Svinøy section in the core of the Norwegian Atlantic Slope Current at position 62°48'N, 4°55'E. Because the current is nearly a barotropic shelf edge current, a single current meter at 100m depth can be used to estimate the total transport of the Norwegian Atlantic Slope Current when scaled with the Svinøy section area (Orvik and Skagseth, 2003a). Applying the scaling factor, the mean Svinøy transport is 4.5 Sv (Table 1).

At the entrance to the Barents Sea, a current meter mooring array has monitored the Atlantic inflow through the Barents Sea Opening since 1997 (Ingvaldsen et al., 2002, 2004). The mooring array extends from 71°30'N to 73°30'N, with the exact number of moorings deployed varying over the measurement period. On average, the Atlantic inflow through the Barents Sea Opening is 2 Sv (Table 1).

## 2.2 ECCOv4-r4 ocean state estimate

To supplement relatively short observational records, we analyse circulation strength in the ECCO Version 4 Release 4 (ECCOv4-r4) ocean state estimate spanning 1992-2017. The ECCOv4-r4 estimate provides a dynamically consistent solution for the global ocean and sea ice state by using nearly all modern ocean observations to constrain an ocean general circulation model with a 1° nominal horizontal resolution (Forget et al., 2015; ECCO Consortium et al., 2021). The ECCOv4-framework



uses the adjoint method to iteratively reduce the model-data misfit by adjusting initial conditions, surface boundary conditions, and model parameters (Heimbach et al., 2005). The observational constraints consist of profiles from Argo floats, Ice-Tethered Profilers, marine mammals, individual CTD stations, as well as satellite observations of sea level, sea surface salinity and temperature, sea ice concentration, and ocean bottom pressure. As observed ocean transport time series are not used as di-

rect constraints in the ECCOv4-framework, the observed and ECCOv4-r4 transport estimates shown here can be considered independent.

Previous releases of the ECCOv4 state estimate have been shown to reproduce well the observed variability in heat and salt in the subpolar North Atlantic (Piecuch et al., 2017; Sanders et al., 2022) and the Nordic Seas (Asbjørnsen et al., 2019; Tesdal and Haine, 2020). In terms of overturning in the North Atlantic, ECCOv4 skilfully reproduces variability at 26.5°N (Evans

et al., 2017; Kostov et al., 2021), though the mean AMOC strength is slightly weaker than in observations (Figure S1). At the OSNAP-East section, the AMOC strength (here in density-space) is also somewhat weaker than in observations (Figure S2). ECCOv4-r4 captures the observed peak in $moc_\sigma$ in 2015/16 (Figure 3), but the observational time series is too short to get a fair assessment of how well interannual variability is represented at OSNAP.

For the upper-ocean components, interannual variability in the Florida Current is very well represented in ECCOv4-r4

(Figure 4a), though the transport magnitude is slightly lower than in observations (27 Sv versus 32 Sv in observations). At the Oleander section, direct comparison in terms of variability is difficult due to the temporal resolution of the Oleander record. However, we note that the Gulf Stream in ECCOv4-r4 fails to intensify sufficiently when moving northward, leaving it too weak at the Gulf Stream separation latitude - a common issue for ocean and climate models (Sen Gupta et al., 2021). The ECCOv4-r4 estimate captures the volume transport magnitude and variability at the Greenland-Scotland Ridge and Svinøy sections well

(Figure 4a). However, the Greenland-Scotland Ridge inflow in ECCOv4-r4 has a too weak Denmark Strait component (0.2 Sv) and too strong Iceland-Faroe Ridge component (4.7 Sv) compared to observations (0.9 Sv and 3.8 Sv in observations, respectively). The climatological mean for the Barents Sea Opening inflow is accurate in ECCOv4-r4 (2.03 Sv in observations vs. 2.16 Sv in ECCOv4-r4), though there is little agreement on interannual variability as discussed previously in Asbjørnsen et al. (2019).

## 2.3 Data treatment


For the observational records with a higher-than-monthly temporal frequency, we compute monthly means of the volume transport time series. This is the case for the Florida Current, Western Boundary Current, RAPID $moc_z$, and Svinøy records. To highlight interannual variability, we filter with a 1-year low-pass triangular filter (24-month filter width; Figure 2). Six months are removed at the start and the end of the filtered transport time series to limit the edge effects from filtering. When filtering,

shorter gaps in the measurement records (no more than five consecutive months) are smoothed over, while more extensive gaps such as the one between November 1998 and May 2000 in the Florida Current record, are treated as a discontinuous time series. Time series are normalized ($\frac{X-\mu_x}{\sigma_x}$) prior to filtering when comparing to the variability in ECCOv4-r4 (Figure 4).

To assess the coherence between the transport time series, we calculate the correlation coefficient between the different low-pass filtered observational and ECCOv4-r4 time series. We show correlation at lag zero (Table 2), and for the ECCOv4-





r4 estimate we also identify the maximum correlation for lag times between zero and six years (Figure S4). Several of the observational time series have a limited overlapping period of available data, and autocorrelation further limits the number of independent data points. We therefore use the Chelton (1983) method at the 95% confidence level to assess whether the correlation between two time series is significant. The method uses the effective degrees of freedom to compute a correlation coefficient value as a threshold for significance. The threshold for significant correlation varies substantially, taking a high

value when the effective degrees of freedom is low.

We estimate linear trends over the extent of the individual observational records using the least squares method (Table 1). Trends in the ECCOv4-r4 state estimate are evaluated over the 1992-2017 period (Table 3). The trend calculations are performed on the unfiltered monthly mean time series. To assess whether the trend values are significantly different from zero at the 95% confidence level, we use the modified Mann-Kendall test for autocorrelated data (Hamed and Rao, 1997).

## 175 3 Observed coherence, variability, and change

Observations of circulation strength at fixed locations are often used to make inferences about the state of the large-scale circulation (e.g. Smeed et al., 2018; Østerhus et al., 2019). Here, we view the different circulation components in the context of the extended Gulf Stream system, focusing on observed coherence between the branches monitored on interannual time scales, and potential trends over the respective measurement periods.

### 180 3.1 Meridional coherence

Circulation in the North Atlantic Ocean adjusts to changes in local surface forcing (wind and buoyancy) through rapid propagation of boundary waves at the western boundary, slow westward propagation of Rossby waves, and advection of density anomalies with the ocean currents (Johnson and Marshall, 2002; Zhang, 2010; Marshall and Johnson, 2013). The range in time scales of these processes communicating change (advection; ∼3-4 years from subpolar to subtropical latitudes, Kelvin waves;

< 1 year, Rossby waves; interannual-decadal time scales) makes the adjustment period potentially long, and the system's meridional coherence is thought to increase with increasing time scale considered (e.g., Gu et al., 2020).

Comparing volume transport time series of the monitored branches show limited meridional coherence within the Gulf Stream system on interannual time scales for all sensible lags (Table 2, Figure S4a). The North Atlantic Current at the OSNAP-East section shows high zero-lag correlation values to the Greenland-Scotland Ridge, Svinøy, and Barents Sea Opening sections

downstream, but the correlations are not statistically significant as the degrees of freedom are low for the short OSNAP record (Table 2). Between the transport at the Svinøy section and the inflow through the Barents Sea Opening, there is a weak but statistically significant relationship (r=0.30) at zero lag time. The remaining observational records show little sign of covariance.

In contrast to the observational records, some more distinct patterns of coherence are found within the ECCOv4-r4 estimate

(Table 2). The strongest relationships identified are at zero lag time between transport sections that are geographically close and upstream of major recirculation branches. Specifically, we find coherence within the Gulf Stream boundary current (Western





**Figure 4.** Interannual to decadal volume transport variability in ECCOv4-r4 and observations. (a) ECCOv4-r4 transport (1992-2017) time series are displayed in colors with corresponding observational time series (as in Figure 2) in black. The time series have been normalized and 1-year low-pass filtered. Significant correlations between observational and ECCOv4-r4 time series are indicated in bold font. (b) Equivalent ECCOv4-r4 transport sections as in (a), but smoothed with a 5-year low-pass filter to highlight decadal variability. The time series in (b) are not normalized so that the magnitude of decadal trends is visible.





**Table 2.** Coherence between the transport sections. Correlations at zero lag time between normalized, low-pass filtered volume transport time series for the observational records (gray cells; see Table 1 for overlapping time periods), and for the equivalent ECCOv4-r4 transport (white cells; 1992-2017 period). For the Oleander observational record, linear interpolation is used to obtain monthly values for the correlation. Significant correlations at the 95% confidence level in bold font (Chelton (1983) method for evaluating significance).

|  | Florida Current | WBC | Oleander GS | OSNAP NAC | GSR | Svinøy | BSO |
|---|---|---|---|---|---|---|---|
| Florida Current | 1 | **0.71** | 0.07 | 0.08 | -0.16 | -0.25 | 0.22 |
| WBC | **0.91** | 1 | 0.26 | -0.16 | 0.21 | 0.15 | 0.25 |
| Oleander GS | **0.71** | **0.78** | 1 | 0.43 | -0.15 | -0.12 | 0.36 |
| OSNAP NAC | -0.27 | -0.20 | -0.19 | 1 | 0.82 | 0.73 | 0.88 |
| GSR | -0.12 | -0.27 | -0.06 | 0.18 | 1 | 0.38 | 0.10 |
| Svinøy | 0.06 | -0.05 | 0.19 | -0.00 | **0.79** | 1 | **0.30** |
| BSO | -0.46 | **-0.63** | -0.54 | 0.09 | **0.52** | **0.32** | 1 |

Boundary Current at 26.5°N and the Oleander section) and within the Nordic Seas (Greenland-Scotland Ridge inflow, Svinøy, Barents Sea Opening). Covariability at zero lag must be a result of fast boundary wave propagation or the ocean responding to regional scale atmospheric forcing. Testing for a range of lag times, we find no covariance between the subtropics and the
subpolar North Atlantic, or the subpolar North Atlantic and the Nordic Seas on interannual time scales in ECCOv4-r4 (Figure S4a).

Within the Ekman layer, the ocean responds to the local surface wind stress independently at each latitude. We therefore additionally check coherence within ECCOv4-r4 when removing the upper 100m before integrating across the sections (Table S1). Removing the Ekman layer does not notably increase the coherence or establish any new relationships between the
analysed sections. Similarly to the full section transports, testing for different lag times reveals no systematic patterns of coherence that can be linked to advection times of anomalies (Figure S4b). We therefore conclude that meridional coherence in the Gulf Stream system is limited to the gyre structures on interannual time scales, considering the Nordic Seas boundary current system as a separate gyre-like structure.

### 3.2 Change over the observational record

Under future emission scenarios, climate models consistently project a weakened AMOC (e.g., Weijer et al., 2020) and to a lesser extent, Gulf Stream (Sen Gupta et al., 2021; Asbjørnsen and Årthun, 2023). There is, however, no consensus on whether such a weakening has already occurred over the past century. Some paleo and proxy reconstructions indicate that the AMOC already has weakened (e.g., Thornalley et al., 2018; Caesar et al., 2021), potentially with as much as 15% since the mid-20th century (Caesar et al., 2018). Kilbourne et al. (2022) argue, on the other hand, that circulation strength from paleo
records is poorly constrained, and advise against concluding from subsets of records. When a more complete set of available





**Table 3.** Linear trends over monthly mean transport time series in ECCOv4-r4 (1992-2017). Significant trends at the 95% confidence level in bold font (modified Mann-Kendall trend test for autocorrelated data (Hamed and Rao, 1997)).

|  | Trend [Sv/yr] |
| --- | --- |
| Florida Current | **-0.0670** |
| RAPID WBC | **-0.0739** |
| Oleander GS | **-0.1078** |
| OSNAP-East NAC | **0.0516** |
| GSR | 0.0061 |
| Svinøy | -0.0119 |
| BSO | **0.0152** |

proxy records for the AMOC is considered the findings are inconclusive (Moffa-Sánchez et al., 2019), illustrating the complex relationship between the ocean state and the different proxy types and locations.

Various methods to reconstruct the circulation strength from historical hydrography or sea level are commonly applied as an alternative to paleo proxies for the most recent century. Fraser and Cunningham (2021) find no statistically significant trend
over the past century (1900-2019) when using the Bernoulli inverse to reconstruct the AMOC strength from hydrography at 50°N. Similarly, Rossby et al. (2020) find no long-term trend (1900-2020) in the reconstructed geostrophic transport of the Nordic Seas inflow, or in the Gulf Stream volume transport from direct observations (Rossby et al., 2014). Using inverse models based on hydrographic transects, Caínzos et al. (2022) find no systematic change in the AMOC at any latitude when comparing the past three decades. Reconstructing the AMOC at 26°N for the 1981-2016 period from hydrography, Worthington et al.
(2021) similarly find no decline in the subtropical AMOC. At the same latitude, Piecuch (2020) finds some indication of a weakening Florida Current over the 1909-2018 period using historical tide gauge measurements and Bayesian analysis.

In the observational records analysed here, only the Florida Current and AMOC at RAPID (both at 26°N) display a statistically significant weakening over their respective observational periods (Table 1). A weakening Florida Current since 1982 is also found in a recent, more comprehensive analysis, combining the cable measurements with altimetry and in-situ measure-
ments and their associated observational uncertainties (Piecuch and Beal, 2023). For the ECCOv4-r4 period (1992-2017), we find a significant weakening trend for all the subtropical sections (Table 3) due to weakening transports between the mid-2000s and mid-2010s (Figure 4b). We note, however, that the weakening trend identified for the subtropical sections cannot be explicitly connected to anthropogenic forcing. Pronounced multidecadal transport variability is highlighted in previous studies (e.g., Fraser and Cunningham, 2021; Rossby et al., 2020) and the 26-year ECCOv4-r4 period is too short to represent such
multidecadal signals. At RAPID, the notable weakening in overturning between 2006 and 2010 is explained by changes in the upper mid-ocean transport and Ekman transport components (Figure S3), which have been shown to result from adjustments to wind forcing (Roberts et al., 2013; Zhao and Johns, 2014). In terms detecting anthropogenically forced weakening at 26°N, as much as 60 years of observations could be required given an observation error of 1 Sv (Baehr et al., 2008).



In the subpolar North Atlantic, the North Atlantic Current at OSNAP-East displays a strengthening after 2007 consistent
with a strengthening subpolar gyre in that period (Koul et al., 2020). At the Greenland-Scotland Ridge and in the Nordic Seas,
the circulation shows no weakening over the different observational records or for the ECCOv4-r4 period (Table 1 & 3). As
pointed out in Østerhus et al. (2019), the observed overflow transports seen in Figure 3 indicate that any AMOC slowdown
during the past two decades does not stem from reduced overturning in the Nordic Seas and Arctic Ocean. North of the
Greenland-Scotland Ridge, transports at Svinøy show no trend (Orvik, 2022, and Table 1). At the Barents Sea Opening there is
a strengthening over the ECCOv4-r4 period which is not seen in the observational record Figure 3b, Table 3). However, trends
in observed sea surface height found in Polyakov et al. (2023) suggest that there might have been an increased transport in the
northernmost Barents Sea Opening inflow branch after the mid-2000s that is not fully captured by the morring array.

Consistent with our results, previous studies have also found differing decadal trends between the subtropical, subpolar,
and Nordic Seas gyres. For instance, Jackson et al. (2022) show evidence of differing decadal trends in the subtropical and
subpolar AMOC over the historical record. They find a strengthening subtropical AMOC from 2001 to 2005 and a weakening
from 2005 to 2014, while the subpolar AMOC likely strengthened from 1980 to the mid-1990s and then weakened until the
2010s. In future emission scenarios, climate models show the Nordic Seas gyre strengthening in the second half of the 21st
century, something which enhances water mass transformation in the Nordic Seas and thus may act as a stabilizing factor for
an overall weakening AMOC south of the Greenland-Scotland Ridge (Årthun et al., 2023).

### 3.3    Mechanisms of interannual to decadal variability

Variability in the North Atlantic and Nordic Seas is closely linked to atmospheric forcing. To identify the atmospheric circu-
lation patterns most closely associated with interannual volume transport variability at the ocean observation sites, we regress
the annual mean sea level pressure onto the annual mean volume transport time series in ECCOv4-r4 (Figure 5, Figure S7) and
in observations (Figure S6). Consistent with the coherence analysis (Table 2), we find interannual variability in the subtropics,
subpolar North Atlantic, and Nordic Seas to be associated with different atmospheric circulation patterns.

For the subtropical ocean transports, a low-pressure anomaly over the Labrador Sea and a basin-wide high-pressure anomaly
over the subtropics are associated with a stronger Gulf Stream on interannual time scales (Figure 5a-b). Previously, Baringer and
Larsen (2001) found a negative correlation between the Florida Current strength and the NAO on interannual time scales, but
the relationship was only seen to hold for the period 1986-1998 (Meinen et al., 2010; Sanchez-Franks et al., 2014). Rather than
the NAO and associated shifts in the latitude and strength of the climatological sea level pressure pattern, Hameed et al. (2021)
find a link between the longitudinal position of the Icelandic Low and Florida Current transport at zero lag time ($r = -0.50$).
When perturbing the ECCOv4-r4 state estimate with the onshore wind stress anomalies associated with an eastward shifted
Icelandic Low, they get a sea level increase along the North American coast and a weakened Florida Current. The pattern
seen in Figure 5a-b associated with a strengthened Gulf Stream resembles a westward shift of the Icelandic Low and is thus
consistent with the mechanism in Hameed et al. (2021). However, we note that several mechanisms not addressed here are
thought to contribute to interannual variability in the Florida Current, such as eddy activity east of Bahamas (Frajka-Williams
et al., 2013), excursions of the Loop Current upstream (Hirschi et al., 2019), and ENSO (Dong et al., 2022).

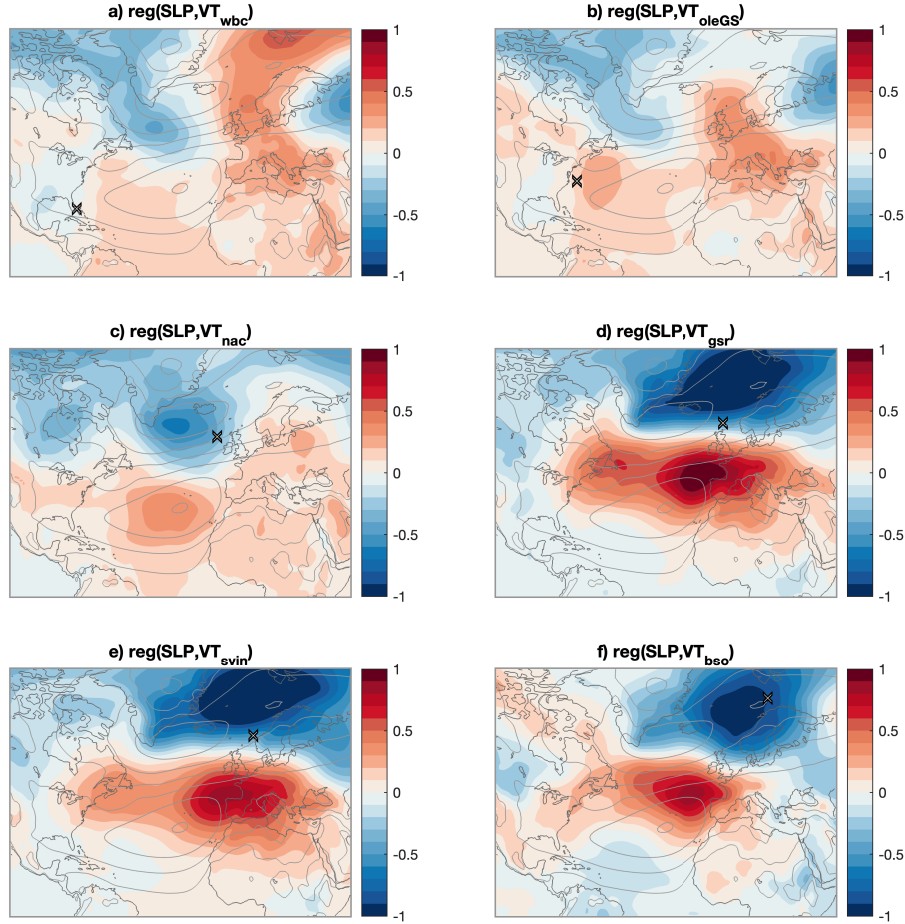

**Figure 5.** Transport variability and large-scale atmospheric circulation patterns. Annual mean sea level pressure (SLP; hPa) regressed onto annual mean volume transport (VT; Sv) time series in ECCOv4-r4; (a) Western Boundary Current at $26.5^{o}$N (wbc), (b) Gulf Stream at Oleander section (oleGS), (c) North Atlantic Current at OSNAP-East (nac), (d) Greenland-Scotland Ridge inflow (gsr), (e) NwASC at Svinøy (svin), and (f) Barents Sea Opening inflow (bso). The volume transport time series has been normalized $\left(\frac{X-\mu_x}{\sigma_x}\right)$ for comparable magnitudes between the panels. Unit is hPa per standard deviation of volume transport. The major features in the regression patterns discussed are significant at the 90% confidence level (Ebisuzaki, 1997). Gray contour lines show the climatological SLP pattern (contour interval: every 3 hPa from 1007 to 1019 hPa). The crosses mark the approximate location for the volume transport time series.





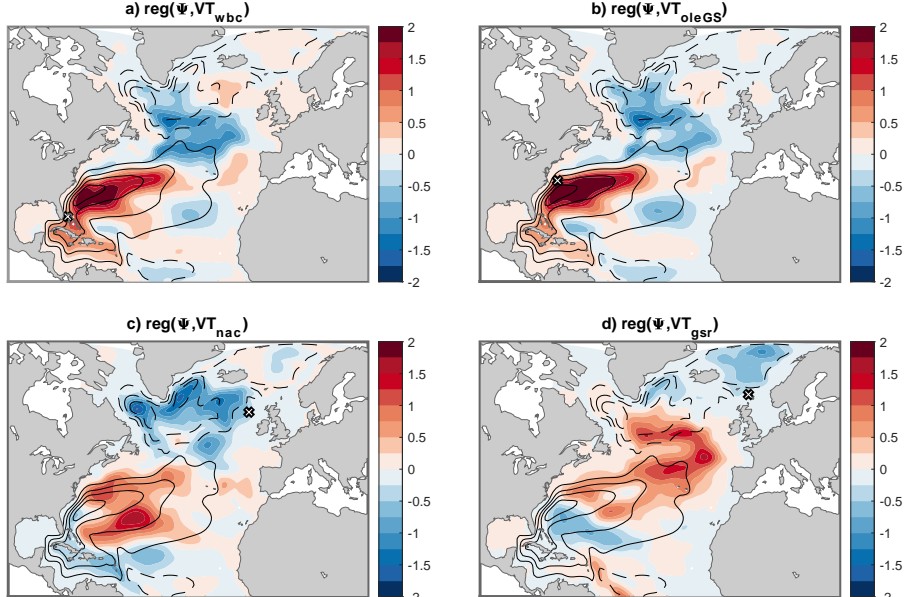

**Figure 6.** Transport variability and large-scale oceanic circulation patterns. Annual mean barotropic streamfunction ($\psi$; Sv) regressed onto annual mean volume transport (VT; Sv) time series in ECCOv4-r4; (a) Western Boundary Current at $26.5^o$N (wbc), (b) Gulf Stream at Oleander section (oleGS), (c) North Atlantic Current at OSNAP-East (nac), (d) Greenland-Scotland Ridge inflow (gsr). The volume transport time series has been normalized ($\frac{X-\mu_x}{\sigma_x}$) for comparable magnitudes between the panels. Unit is Sv per standard deviation of volume transport. Gray contour lines show the climatological barotropic streamfunction pattern (dashed line where $\psi$ takes negative values). The crosses mark the approximate location for the volume transport time series. The major features in the regression patterns discussed are significant at the 90% confidence level (Ebisuzaki, 1997).

For the North Atlantic Current across OSNAP-East, an increased transport is associated with a strengthened climatological sea level pressure pattern resembling the NAO in a positive state (NAO+; Figure 5c). On interannual time scales, the pattern

likely relates to locally strengthened westerly winds which strengthens the subpolar gyre as seen in the associated barotropic streamfunction in Figure 6c. In ECCOv4-r4, a strengthened North Atlantic Current across OSNAP-East thus reflects a stronger subpolar gyre, but does not necessarily lead to a strengthened inflow across the Greenland-Scotland Ridge or a strengthened Nordic Seas gyre. The finding is consistent with the low correlations between the North Atlantic Current and the downstream components in Table 2. The decadal trends in the North Atlantic Current (Figure 4b) agree with multiple subpolar gyre indecies

indicating a weakening subpolar gyre from the mid-1990s to 2005, followed by a strengthening (Koul et al., 2020). The evolution fits with the accumulated historic NAO forcing seen in Figure S6, consistent with persistent NAO+ conditions spinning up the subpolar gyre due to strengthened wind stress curl and elevated heat loss (Eden and Willebrand, 2001; Sarafanov, 2009).

The sea level pressure pattern associated with a strong Greenland-Scotland Ridge inflow (Figure 5d) mainly arises from the Faroe-Shetland Channel component which dominates the net inflow variability (Figure S3). The pattern in Figure 5d shows

a strengthened and northeastward shifted Icelandic low and Azores high, which is near identical to the equivalent regressions



for the Svinøy and Barents Sea Opening sections (Figure 5e-f). The northeastward shift suggests a corresponding shift of the westerlies and the storm tracks. From previous studies it is well established that the Faroe-Shetland Channel inflow typically increase under NAO+ conditions due to a strengthened sea surface height gradient across the channel (e.g. Chafik, 2012; Bringedal et al., 2018). Regressing ERA5 sea level pressure onto the observational Faroe-Shetland Channel volume transport

shows a more canonical NAO+ anomaly consistent with previous studies (not shown). For variability at the Svinøy section, the relationship with the NAO is less straightforward and known to be more closely associated with the position of the westerlies rather than the strength (Orvik, 2022). More low-pressure systems directed into the Nordic Seas due to a northeastward shifted storm track strengthen the southwesterly winds along the Norwegian coast, and sets up onshore Ekman transport and piling along the coast which in turn strengthens the Norwegian Atlantic Slope Current (Skagseth and Orvik, 2002; Richter et al.,

2009). The barotropic streamfunction anomaly associated with a stronger Greenland-Scotland Ridge inflow (and Svinøy and Barents Sea Opening components) is a strengthened Nordic Seas gyre (Figure 6d). Moreover, the anticyclonic anomaly in the intergyre-region seen in Figure 6d can be interpreted as a more tilted North Atlantic Current (Marshall et al., 2001), which potentially means that more water crosses the Greenland-Scotland Ridge and less recirculates within the subpolar gyre.

The barotropic streamfunction anomaly patterns seen in Figure 6 indicate that strong transports at the individual sections are

typically associated with a strengthened gyre structure locally, with little sign of the other two gyres strengthening simultaneously. This suggests that recirculation and branching within the three major gyre structures is likely a key factor in explaining the lack of coherence between the gyres. For instance, downstream of the 95 Sv in the Gulf Stream core at Oleander, substantial subtropical recirculation occurs (Mann, 1967; Meinen and Watts, 2000) as well as mixing with subpolar water masses (e.g., Brambilla et al., 2008) before the North Atlantic Current transports roughly 20 Sv across OSNAP-East. Of the 20 Sv crossing

OSNAP-East, only 7-8 Sv crosses the Greenland-Scotland Ridge meaning that roughly 50% of the water recirculates within the subpolar gyre (Table 1). Our results thus indicate that while the subtropical gyre, subpolar gyre, and Nordic Seas gyre are connected through the northward transport of subtropical-origin water, they are disconnected by the recirculation within the gyres (Figure 7).

## 4   Summary and conclusions

In this study, we have synthesized available ocean transport measurements and the ECCOv4-r4 ocean state estimate to investigate variability within the Gulf Stream system on interannual to decadal time scales. We find little coherence between the observational records at different latitudes on interannual time scales (Table 2). In the ECCOv4-r4 estimate we find evidence of regional coherence, with subtropical variability being distinct from subpolar and Nordic Seas variability. These findings also translate to decadal time scales, where we in ECCOv4-r4 find a weakening Florida Current at $26.5^{o}$N and Gulf Stream at the

Oleander section after the mid-2000s, while the Nordic Seas inflow and circulation remained stable or strengthened (Table 3, Figure 4b).

A higher degree of coherence within the ECCOv4-r4 framework compared to the observational records can be due to a number of reasons. Firstly, the overlapping time periods between some of the observational records are short, making the





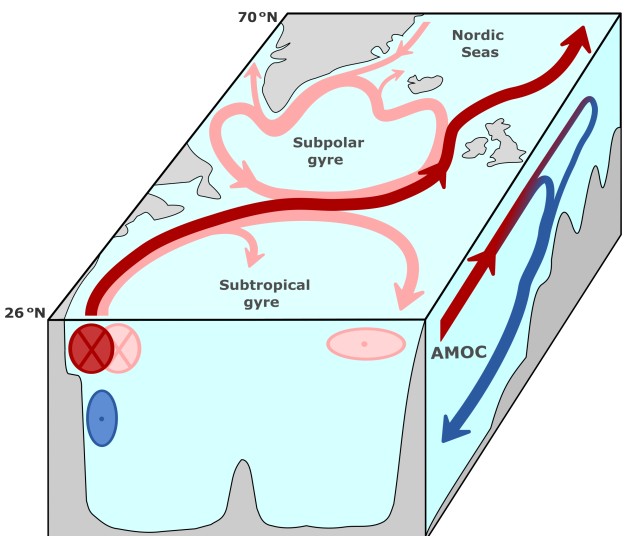

**Figure 7.** Idealized view of the North Atlantic and Nordic Seas circulation. The Gulf Stream system is connected through the northward transport of subtropical-origin water (red color), but disconnected by recirculation within the gyre structures (pink color). Seen from a mass-balance perspective at $26^{o}$N, the Gulf Stream is shown using two circles highlighting it partly compensating for the Deep Western Boundary Current and partly being the western boundary of the wind-driven subtropical gyre. The Deep Western Boundary Current flowing south is represented by the blue circle at depth. The zonally-integrated view of the circulation is shown on the side, illustrating warm upper-ocean water being gradually transformed and sinking at high latitudes as a part of the AMOC.

threshold for significance high. Secondly, ECCOv4-r4 has a coarse model grid which will smooth high-frequency variability
from, for instance, eddies. Thirdly, just as models have their biases, observational records have observation errors related
to calibration, sampling, and system design (e.g., McCarthy et al., 2015) which potentially could hide a more meridionally
coherent signal.

The limited coherence across the subtropical, subpolar, and Nordic Seas gyres identified here highlights the role of local
oceanic response to atmospheric circulation patterns. Specifically, transport variability within the gyres is associated with vari-
ability in the position and strength of the Azores high and the Icelandic low (Figure 5), meaning that regional atmospheric
circulation patterns are a major influence at the observation locations through strengthening or weakening the different gyres.
Removing the Ekman layer transports from the analysis does not notably increase covariance between the sections on inter-
annual time scales (Table S1), meaning that it is not simply transport anomalies in the Ekman layer that overshadow a more
meridionally coherent signal. While we focus on upper-ocean transports in the Gulf Stream system, our findings agree with
observed (Lozier et al., 2010; Frajka-Williams et al., 2019; Moat et al., 2020; Jackson et al., 2022) and simulated (Bingham
et al., 2007; Gu et al., 2020) patterns of disconnect between subpolar and subtropical AMOC, highlighting different overturning
behaviours between the gyres (Figure 7).




The Gulf Stream is projected to weaken under continued high emissions, both as a consequence of a weakened Deep Western Boundary Current and a weakened subtropical gyre circulation (Beadling et al., 2018; Asbjørnsen and Årthun, 2023). Being limited to interannual-decadal time scales, we are unable to determine how the observational trends over the respective measurement periods (Table 1) relate to anthropongenic forcing. We note, however, that none of the circulation branches display any signs of past or near-future collapse. The RAPID record, moreover, shows that sizable shorter term trends such as the reduced overturning between 2006 and 2010 (Figure S3) can occur from oceanic adjustments to surface wind forcing (Roberts et al., 2013; Zhao and Johns, 2014; Kostov et al., 2021).

In finding little coherence between the gyre structures on interannual to decadal time scales, our results reinforce the need for caution in inferring large-scale circulation change from single observational records within the time scales that are currently resolved. Improved mechanistic understanding of the variability and continued monitoring of the circulation at a range of latitudes is therefore required to predict and detect emerging trends.

*Data availability.*   The ECCOv4-r4 ocean state estimate (ECCO Consortium et al., 2021) is available at https://ecco-group.org/products.htm. ERA5 reanalysis data (Hersbach et al., 2020) is available at https://doi.org/10.24381/cds.f17050d7. Observational Barents Sea Opening and Svinøy volume transport time series (Ingvaldsen et al., 2004; Orvik, 2022) are available through the Norwegian Marine Data Centre (http://metadata.nmdc.no/UserInterface) and are provided by the Institute of Marine Research and University of Bergen, respectively. Greenland-Scotland Ridge volume transports (Østerhus et al., 2019) are available online at http://www.oceansites.org/tma/gsr.html. The OS-NAP observational data (Fu et al., 2023) is available at https://doi.org/10.35090/gatech/70342 through the Overturning in the Subpolar North Atlantic Program. Oleander section volume fluxes (Rossby et al., 2019) are available at https://oleander.bios.asu.edu/data/oleander-fluxes/ through the Oleander Project. The RAPID-MOCHA-WBTS observational data (Moat et al., 2022) is available at https://doi.org/10.5285/e91b10af-6f0a-7fa7-e053-6c86abc05a09 through the RAPID-Meridional Overturning Circulation and Heatflux Array-Western Boundary Time Series programme. The Florida Current volume transports (Meinen et al., 2010) are available on the Atlantic Oceanographic and Meteorological Laboratory web page (www.aoml.noaa.gov/phod/floridacurrent/) through the DOC-NOAA Climate Program Office - Ocean Observing and Monitoring Division.

*Author contributions.*   HA, TE, and HLJ defined the overall research problem and methodology. HA carried out the analysis and led the writing of the manuscript. JS developed the conceptual figure (Figure 7) and contributed to Figure 1. All co-authors discussed the analysis, refined the methodology, and contributed to the text.

*Competing interests.*   The contact author has declared that none of the authors has any competing interests.



*Acknowledgements.* We acknowledge the teams behind the RAPID-MOCHA-WBTS projects, Oleander Project, OSNAP program, AtlantOS project, and the Norwegian Institute of Marine Research for collecting and making data freely available. We thank David Smeed and Tom Rossby for kindly sharing updated records for the Western Boundary Current at $26.5^o$N and the Oleander section, respectively. We also thank Kjell Arild Orvik and Algot Pedersen for sharing their knowledge about the Svinøy record and the most recent data. HLJ is grateful for funding from the NERC-NSF SNAP-DRAGON project (Grant NE/T013494/1).



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
