# Peer review of "Observed change and the extent of coherence in the Gulf Stream system"

_EGUsphere, 2023_

## Referee Comment (RC2)

**Comments on "Observed change and the extent of coherence in the Gulf Stream system" by Asbjørnsen et al.**

In this manuscript, the authors present new evidence of a lack of coherence between the subtropical gyre, subpolar gyre and Nordic Seas in the North Atlantic, while also discussing possible trends in the strength of the circulation. They take advantage of the observational data available between 24N and 75N, also relying on models. Instead of focusing on AMOC transports, the authors have provided results using components of the Gulf Stream System along the three gyre structures. Moreover, they propose mechanisms of interannual to decadal variability linked to atmospheric forcing.

The authors have presented a well written paper with a clear methodology. Their new approach using certain components of the upper ocean circulation instead of the AMOC integrated view is of high interest for understanding the meridional coherence of the North Atlantic circulation.

As a result, I find this manuscript should be suitable for publication after addressing some minor comments given below.
* * *
Firstly, I recommend the authors to emphasize the main point of the results – gyre-specific meridional coherence, specially between SPNA and Nordic Seas. This disconnection between subtropics and subpolar gyres is not that new, but it is interesting to see these differences with the Nordic Seas, considering the great effort on providing the GSR, Svinøy and BSO time series.

Therefore, I would like to see a more extended discussion on gyre-specific coherence. On this topic, Buckley & Marshall (2016) in their review state that:

'Modeling studies [*Bingham et al.*, 2007], ocean state estimates [*Wunsch and Heimbach*, 2013b], and observations [*Mielke et al.*, 2013] indicate that the AMOC is not coherent between the subtropical and subpolar gyres on interannual timescales. Within the subtropical gyre interannual AMOC variability is dominant, while in the subpolar latitudes decadal AMOC variability is stronger [*Balmaseda et al.*, 2007; *Wunsch and Heimbach*, 2013b]. On decadal timescales models and state estimates generally exhibit meridionally coherent modes of AMOC variability.'

More recently, Zhao (2018) relay the importance of mesoscale processes on transporting MHT poleward across the SPNA using models. Zou (2019), similarly to this manuscript, investigate the coherence in the North Atlantic in deep layers through the equatorward NADW rather than AMOC and in Zou (2020), they re-examine the meridional structure of AMOC variability and diagnoses the associated forcing scenarios with three different models, showing that AMOC variability south of the Labrador Sea can be decomposed into a latitudinally coherent component and a gyre-opposing component, with different variabilities and forcing affecting each. And Han (2023) studies AMOC connection between OSNAP and RAPID in adiabatic terms using numerical models, where the Labrador Sea plays an important role as the origin of that adiabatic forcing that generates the SPNA variability.

Finally, I encourage the authors to add uncertainties to the values computed. Even if we know it is significant with a statistical test, knowing the uncertainties can help us interpret the results (such as the average mean value and trends for both observations and ECCO).

*SPECIFIC COMMENTS*

- Line 61. This is the first reference to the Norwegian Atlantic Current. Throughout the manuscript (text, figures and tables) there are references to both this and the Norwegian Atlantic Slope Current. I think it would be beneficial if the authors clarified both currents and maybe unified them in only just one.
- Table 1 provides a lot of information, but I find it could be useful if some of it were provided later in the manuscript.
  - For example, the trends are mostly discussed later, along with Table 3. The authors should consider moving these trends for observations to Table 3, so that it would be easier to follow the discussion in the text.
  - Please specify the uncertainties of the mean (or standard deviation of the mean), even after marking significant trends, as it provides an idea of the variability of the dataset.
  - Some of the naming don't follow the same structure (RAPID WBC and RAPID MOCz but GSR and GSR OW; Oleander GS but Svinøy) – do the authors want to specify the current part of the Gulf Stream System?
  - The values in Table 1 for mean and trend are not always reported in the literature cited. 'Data source' would be a more appropriate term than 'reference' to cite the works from which the datasets were obtained.
  - One of the points that I have not seen specified along the manuscript is the sensitivity of these computations on how the authors have defined the different currents studied, i.e., which are the horizontal and vertical boundaries of each current and what criteria were the authors following.
  - On another note, the instrumentation and methodology used for each monitoring observing system is quite diverse. The authors could consider adding columns for the instrumentation used for each timeseries and the frequency of the observations.
- Lines 98-99: 'the Oleander record has different temporal resolution than the other time series displayed in Figure 2'. This is the first mention of the time resolution of the time series, so to make this statement the authors should include that information when describing the rest of time series. This is related to the beginning of section 2.3, where it is stated that 'For the observational records with a higher-than-monthly temporal frequency…' and then they specify which ones those are. It can be a bit confusing, so I recommend specifying when describing each dataset.
- Figure 3. This is up to the authors, but I encourage them to add figure 3 as another column to figure 2, so that it will be referenced to the positions in the map and comparable to the individual currents.
  - For the GSR, there is a sharp difference in the behaviour of the overflow between the two intervals: could the authors briefly describe why is that? Are there any difference in instrumentation or methodology on all or any of the sections included in GSR? Why are there two intervals for the overflow (Figure 3) and not for the current (Figure 2)?
- Lines 107-115. This paragraph is about the GSR and each component. The description of IFR, FSC and DS can be complemented with a call for Figure S3a to show that the anomalies respond to that of FSC, even if the mean transport is larger over IFR.
- Line 162. The authors have normalized the datasets, which can be useful when comparing variables with different units. Could the authors discuss briefly this choice instead of just computing anomalies?
- Lines 218-226. This is a good paragraph discussing reconstructions of AMOC with observations. When discussing inverse models, the recent paper by Fu et al., (2020) could be included, where the authors find no trends in AMOC creating boxes between 24N and 55N.
- Table 3. As stated above, even if we don't have trends here for AMOC from ECCO, this is a better place for trends than Table 1. Also, these trends should be expressed with their uncertainties.

- Lines 317-322: This is a good discussion on the comparison between ECCO and observations. However, it has not been mentioned previously in the manuscript, so the authors may consider placing it above and not in the 'Summary and conclusions' section. One opportunity could be between lines 201 and 202, after discussing that ECCO finds more patterns than observations and before discussing the Ekman layer.

*TECHNICAL CORRECTIONS*

- Throughout the manuscript, please make sure that the main currents cited and discussed in the text are defined at their first mention and at the appropriate figures and tables.
- Please make sure that the figures in Supplementary Information are in order of appearance in the main text.
- Figure 1. I find this figure useful to illustrate the introduction, but the authors could include some extra details to make it more accessible.
    - On the schematic map in A, only the Gulf Stream is specified with its full name. However, that is not the case for DWBC, EGC, NAC and NwASC. I understand there's not much space in the figure, but it would be useful to define the acronyms at least in the figure caption. NAC, EGC and DWBC are common enough, but that's not the case for NwASC.
    - I suggest adding the A16 cruise track from subplot B to the map in A.
    - On subplot B, it could be helpful to employ the same colours for the arrows representing the upper (purple) and deep (black) circulation.
- Line 34. 'of which the Gulf Stream and **the** extensions' changed to 'of which the Gulf Stream and **its** extensions'
- Lines 53-54: the authors could describe shortly the location of the RAPID and OSNAP array (subtropical and subpolar or 24N and 55N). E.g.: The AMOC strength has been measured by cross-basin observing systems at 24N since 2004 (RAPID; Cunningham et al., 2007) and at 55N since 2014 (OSNAP; Lozier et al., 2017).
- Line 79. Please specify: 'The **mean** 32 Sv transported by the Florida Current and the, on average, 4.7 **± 7.5** Sv in the Antilles Current …'
- Figure 2. It could be very helpful to include the name of the observing systems in the map, with the color legend applied for the time series, even if it were just the acronyms and they were defined in the figure caption.
- Line 119: 'a single current meter at **100m** depth' is missing a space in '**100 m**'.
- Line 142: 'ECCOv4-r4 captures the observed peak in moc$_\sigma$ in 2015/16 (Figure 3), but the observational time series is too short to get a fair assessment of how well interannual variability is represented at OSNAP.' I understand that this sentence refers to the peak in MOC observations from Figure 3, but it turns out a bit confusing, as there's no ECCO values to compare it against. I would refer readers only to Figure S2c, where the authors can specify the short overlap between both time series (2014-2017).
- Lines 151-152: 'compared to observations (0.9 Sv and 3.8 Sv **in observations**, respectively)'. The second 'in observations' is redundant: 'compared to observations (0.9 Sv and 3.8 Sv, respectively)'.
- Figure 4. The y-axis labels should include magnitude and unit following the same structure as before: a) STD and b) VT [Sv].
- Line 165. Reference to Figure S4 appears before Figure S3.
- Line 236. Reference to Figure S3c instead of only Figure S3.
- Line 258-259: 'onto the annual mean volume transport time series in ECCOv4-r4 (Figure 5, **Figure S7**) and

- in observations (**Figure S6**)'. I think the references for the supplementary information figures are wrong: 'series in ECCOv4-r4 (Figure 5, **Figure S8**) and in observations (**Figure S7**)'.
- Line 279: '**indecies**' changed to '**indices**'.

**References for this review:**

Buckley, M. W., & Marshall, J. (2016). Observations, inferences, and mechanisms of the Atlantic Meridional Overturning Circulation: A review. *Reviews of Geophysics*, *54*(1), 5–63. https://doi.org/10.1002/2015RG000493

Fu, Y., Li, F., Karstensen, J., & Wang, C. (2020). A stable Atlantic Meridional Overturning Circulation in a changing North Atlantic Ocean since the 1990s. *Science Advances*, *6*(48), eabc7836. https://doi.org/10.1126/sciadv.abc7836

Han, L. (2023). Exploring the AMOC Connectivity Between the RAPID and OSNAP Lines With a Model-Based Data Set. *Geophysical Research Letters*, *50*(19), 1–10. https://doi.org/10.1029/2023GL105225

Zhao, J., Bower, A., Yang, J., Lin, X., & Holliday, N. P. (2018). Meridional heat transport variability induced by mesoscale processes in the subpolar North Atlantic. *Nature Communications*, *9*. https://doi.org/10.1038/s41467-018-03134-x

Zou, S., Lozier, M. S., & Buckley, M. (2019). How Is Meridional Coherence Maintained in the Lower Limb of the Atlantic Meridional Overturning Circulation? *Geophysical Research Letters*, *46*(1), 244–252. https://doi.org/10.1029/2018GL080958

Zou, S., Lozier, M. S., & Xu, X. (2020). Latitudinal structure of the meridional overturning circulation variability on interannual to decadal time scales in the North Atlantic Ocean. *Journal of Climate*, *33*(9), 3845–3862. https://doi.org/10.1175/jcli-d-19-0215.1

---

## Author Comment (AC1)

**Reviewer 1**

The authors use available data and model products to quantify correlation between different components of the Gulf Stream system during the past couple decades. The main take-home message is that, during recent interannual and decadal periods, the system shows gyre-scale structure: subtropical circulation features—Florida Current, western boundary current, Oleander Gulf Stream— are correlated with one another to varying degrees, but uncorrelated with elements in the Nordic Seas—Barents Sea Opening, Svinøy, Greenland-Scotland Ridge inflow. The authors also identify potential forcing mechanisms related to large-scale modes of surface climate variation as well as whether these circulation features exhibit significant trends.

This is an excellent paper. The authors study an important question relevant to large-scale observing systems of the North Atlantic. By focusing on particular flow features within the Gulf Stream system, rather than overturning streamfunction, the authors provide a valuable new perspective on the meridional coherence of North Atlantic Ocean circulation—a topic of longstanding interest. The paper is well-written and clear, the reasoning is logical, and the conclusions follow naturally from the results. From what I can tell, the methodology and analysis are scientifically sound.

I found very little (if anything) to criticize here. My minor comments are given below. The paper should be suitable for publication after minor revisions.

Congratulations to the authors on a very nice study

Best,
Chris Piecuch, Woods Hole

*We thank the reviewer, Chris Piecuch, for constructive comments and encouraging feedback. Please find our response below to each of the points raised.*

* References to "the Bahamas" (lower-case "t") should be changed to "The Bahamas" (upper-case "T")

*We have changed to the correct capital 'T' accordingly.*

* I found this sentence near lines ~90-95 confusing: "along a transect from New Jersey to Bermuda (Rossby et al., 2005). The ADCP measurements reach 250-400m depth for the 1992-2004 period, and 500-600m from 2005 and onwards (Sanchez-Franks et al., 2014)." To me, it reads like the ADCP measurements are made *over the range* 250-400 m for 1992-2004 and *over the range* 500-600 m from 2005 onwards. Rather, I think what the authors mean is that the ADCP measures velocities *down to* 250-400 m and *down to* 500-600 m from 2005 onwards. I suggest to clarify.

*As suggested by the reviewer, we have altered the sentence so that it is clear the ADCP measures velocities down to the depth ranges stated (l.92-93).*

* ~Line 134 "... ocean bottom pressure FROM GRACE AND GRACE-FO"

*We have added this information to l.136.*

* I think the figure reference on line 142 should be to Figure 4?

*Thanks for pointing out this imprecision – it is more accurate to refer to Figure S4c here (l.151).*

* It'd be informative to explain how the authors compute "Florida Current Transport" in ECCOv4-r4 and how thy distinguish it from the total western boundary current transport. The resolution of ECCOv4-r4 is very coarse, and the model bathymetry in that region is heavily modified, such that the

"Florida Straits" in the model are much broader than in reality, and the depiction of The Bahamas very unrealistic.

*As the reviewer states, the coarse ECCO-grid does not fully resolve the complex topography in e.g. the Straits of Florida. We have added a paragraph in the methods section explaining the process in defining currents in ECCOv4-r4 that are comparable to the observed currents (l.139-145). We explicitly state how the Florida Current and the Western Boundary Current at 26.5N were defined on the ECCO-grid. We have also added a section to the Supplementary detailing all six transports section definitions in ECCOv4-r4.*

\* Can the authors explain why they normalize the ECCOv4-r4 and observational time series for comparison? I think the result would be more powerful if the authors didn't divide by the standard deviation.

*Based on this comment and similar comments from Reviewer 2, we have chosen to alter Figure 4 so that panel a) shows the volume transport anomalies from the time mean instead of the normalized volume transport. We also find that this makes for a more intuitive comparison with ECCOv4-4.*

\* Minor point on ~Line 205 and Table S1. I suggest to place the correlation values in the lower-left part of the matrix rather than the upper-right so that it's easier for the reader to go back and forth comparing to the corresponding values in Table 2.

*Following the reviewer's suggestion, the correlations in supplementary table Table S1 have been moved to the lower-left corner accordingly.*

\* ~Line 235 the authors should also cite work by Lobelle et al. on this point (https://agupubs.onlinelibrary.wiley.com/doi/full/10.1029/2020GL089974)

*A reference to Lobelle et al. 2020 has been included in l.262.*

\* Line 241 table -> tables

**This has been corrected.**

\* Line 259ff and Figure 5. The authors' analysis here, being based on correlation coefficients, only considers in-phase or anti-phase relationships between transport and sea-level pressure. But earlier, the authors talked about the various mechanisms that mediate the ocean response, which include processes that impart lags and phase differences between forcing and response. Are the authors confident that figures like Figure 5 entirely capture relationships between transport and sea-level pressure? It may be informative also to consider the relationship between transport and the Hilbert transform of sea-level pressure.

*In Figure 5, we shed light on one well-known driver of ocean variability (atmospheric circulation represented by sea level pressure), which at zero-lag serves as a partial explanation for what the current measurements show. The reviewer makes an important point in that the ocean response to atmospheric circulation is complex and occurs on a range of time scales. Additionally, the ocean influences the atmosphere through SST feedback. We have added the following paragraph to be clearer on the interpretation of, and limitations to, the regression analysis shown (l.325-329): "We do find relatively straightforward relationships between regional atmospheric circulation (represented by sea level pressure) and the section volume transports. These zero-lag regressions (Figure 5) are likely most representative of sea level pressure patterns related to ocean circulation's relative immediate barotropic response to anomalous atmospheric forcing (e.g., Eden & Willebrand 2001). It should be noted that the ocean responds to the atmosphere on a range of time scales and also influences the atmosphere through feedback mechanisms (Marshall et al. 2001)."*

* Line 259 and elsewhere (e.g., Table 2). The authors refer to their analysis as measuring "coherence" between different quantities. This isn't strictly true. Coherence is a measure in the frequency domain. I suggest the authors adjust the language to more clearly say theirs is a correlation analysis not a coherence analysis.

*We acknowledge the reviewer's note on the appropriate use of terminology. However, in common language and in scientific literature (e.g., Bingham et al. 2007, Gu et al. 2020, Frajka-Williams et al. 2023), a similar use to ours is typical. As a middle ground, we have partly remained by our use but also elaborated on the usage, hopefully in line with the reviewer's suggestion. Specifically, we have added a sentence explaining our non-statistical use of the expression 'meridional coherence' in the methods section (l.179-180) and altered the sentences mentioning 'coherence' prior to the explanation (e.g., l.6, l.39, l.63). We have also changed the word 'coherence' to 'correlation' in the table captions for Table 2 and Table S1.*

* Line 279. indecies -> indices

*This has been corrected.*

* Discussion section. The paper focuses on volume transports. But, with the exception of coastal sea level, what we really care about from a climate perspective is the transport of heat and other tracers like carbon. Based on their results, can the authors briefly speculate on the potential meridional coherence of heat transport or transport of other tracers?

*The reviewer raises a very valid and intriguing point. We have added a paragraph to the discussion section putting our results in the context of poleward propagation of heat anomalies (l.364-367): "While observations and models show that ocean heat anomalies and other tracers can propagate persistently poleward through the North Atlantic Ocean, leading to potential for skillful climate prediction (e.g, Keenlyside et al. 2008, Årthun et al. 2017), our results herein show that volume transport anomalies do not. Therefore, the mechanism by which the gyres exchange, for instance, heat anomalies, remains unclear and is thus a challenge to address following up on the present study."*

---

## Author Comment (AC2)

**Reviewer 2**

In this manuscript, the authors present new evidence of a lack of coherence between the subtropical gyre, subpolar gyre and Nordic Seas in the North Atlantic, while also discussing possible trends in the strength of the circulation. They take advantage of the observational data available between 24N and 75N, also relying on models. Instead of focusing on AMOC transports, the authors have provided results using components of the Gulf Stream System along the three gyre structures. Moreover, they propose mechanisms of interannual to decadal variability linked to atmospheric forcing.

The authors have presented a well written paper with a clear methodology. Their new approach using certain components of the upper ocean circulation instead of the AMOC integrated view is of high interest for understanding the meridional coherence of the North Atlantic circulation.

As a result, I find this manuscript should be suitable for publication after addressing some minor comments given below.

*We thank the reviewer for insightful comments and suggestions. In line with the reviewer's comments, we have added more details to the discussion of meridional coherence in the AMOC, including a number of the references suggested by the reviewer. We have also made several smaller adjustments to figures, tables, and descriptions based on the reviewer feedback. Please find our detailed response below to each of the points raised.*

Firstly, I recommend the authors to emphasize the main point of the results – gyre-specific meridional coherence, specially between SPNA and Nordic Seas. This disconnection between subtropics and subpolar gyres is not that new, but it is interesting to see these differences with the Nordic Seas, considering the great effort on providing the GSR, Svinøy and BSO time series.

*The gyre-specific meridional coherence we find based on the correlations in Table 2 is within the subtropical gyre and the Nordic Seas, respectively (see e.g. abstract and l.216-222). Following the reviewer's suggestion we have further emphasized this in l.229-232. We also repeat the point of meridional coherence restricted to the gyre structures in Section 4 as a main conclusion (l.341-347, l.354-363).*

Therefore, I would like to see a more extended discussion on gyre-specific coherence. On this topic, Buckley & Marshall (2016) in their review state that: *'Modeling studies [Bingham et al., 2007], ocean state estimates [Wunsch and Heimbach, 2013b], and observations [Mielke et al., 2013] indicate that the AMOC is not coherent between the subtropical and subpolar gyres on interannual timescales. Within the subtropical gyre interannual AMOC variability is dominant, while in the subpolar latitudes decadal AMOC variability is stronger [Balmaseda et al., 2007; Wunsch and Heimbach, 2013b]. On decadal timescales models and state estimates generally exhibit meridionally coherent modes of AMOC variability.'*

*See the response to the comment below for details on how we have extended the discussion of previous literature. As the reviewer points out, lacking meridional coherence between subtropical and subpolar latitudes on interannual time scales is well-known from literature. Previous literature focus on the AMOC (which is an integrated quantity), not on the upper-ocean branches we choose to investigate here. However, we do consider discussing our findings in the context of this literature very important.*

More recently, Zhao (2018) relay the importance of mesoscale processes on transporting MHT poleward across the SPNA using models. Zou (2019), similarly to this manuscript, investigate the coherence in the North Atlantic in deep layers through the equatorward NADW rather than AMOC and in Zou (2020), they re-examine the meridional structure of AMOC variability and diagnoses the associated forcing scenarios with three different models, showing that AMOC variability south of the Labrador Sea can be decomposed into a latitudinally coherent component and a gyre-opposing

component, with different variabilities and forcing affecting each. And Han (2023) studies AMOC connection between OSNAP and RAPID in adiabatic terms using numerical models, where the Labrador Sea plays an important role as the origin of that adiabatic forcing that generates the SPNA variability.

*We have extended the discussion of previous literature investigating limited meridional coherence between subtropical AMOC and subpolar AMOC in Section 3.1 (l.197-206), including several of the references mentioned by the reviewer. We also introduce the concept of limited observed and simulated meridional coherence in the AMOC in Section 1.1 (l.38-42), and come back to it in the summary and conclusion (l.360-363).*

Finally, I encourage the authors to add uncertainties to the values computed. Even if we know it is significant with a statistical test, knowing the uncertainties can help us interpret the results (such as the average mean value and trends for both observations and ECCO).

*We have added the standard deviation of the monthly mean transports to Table 1 and to the descriptions of the measurements in Section 2.1. We have not included full uncertainty estimates of the observed mean transports as these would include uncertainties from instrumental precision, calibration, and sampling. The exact methods for providing such uncertainty estimates likely vary between the groups responsible for the different observing systems and the numbers might not be comparable in a meaningful way. We have added a figure to the supplementary (Figure S6) to show the 95% confidence interval for the estimated trends in observations and in ECCOv4-r4.*

SPECIFIC COMMENTS

• Line 61. This is the first reference to the Norwegian Atlantic Current. Throughout the manuscript (text, figures and tables) there are references to both this and the Norwegian Atlantic Slope Current. I think it would be beneficial if the authors clarified both currents and maybe unified them in only just one.

*We have added sentences to the manuscript when introducing the Svinøy transect measurements of the Norwegian Atlantic Slope Current (l.117-118), describing how the Norwegian Atlantic Current is a two-branch current system; the Norwegian Atlantic Slope Current following the continental shelf and the Norwegian Atlantic Front Current further offshore.*

• Table 1 provides a lot of information, but I find it could be useful if some of it were provided later in the manuscript.

- For example, the trends are mostly discussed later, along with Table 3. The authors should consider moving these trends for observations to Table 3, so that it would be easier to follow the discussion in the text.

    *We have removed the observed trends from Table 1 and added them to Table 3. The trends in the observed overturning components have been moved to the supplementary (Table S2).*

- Please specify the uncertainties of the mean (or standard deviation of the mean), even after marking significant trends, as it provides an idea of the variability of the dataset.

    *As the reviewer suggests, we have added the standard deviation of the monthly means to Table 1 and to the descriptions of the measurements in Section 2.1.*

- Some of the naming don't follow the same structure (RAPID WBC and RAPID MOCz but GSR and GSR OW; Oleander GS but Svinøy) – do the authors want to specify the current part of the Gulf Stream System?

*We have changed the legends of Figure 2-4 so that they both state the section name and the abbreviation for the current/component shown. For consistency, we have also done the same for Table 1-3.*

- The values in Table 1 for mean and trend are not always reported in the literature cited. 'Data source' would be a more appropriate term than 'reference' to cite the works from which the datasets were obtained.

*We have changed the header from 'Reference' to 'Data source' in Table 1 accordingly.*

- One of the points that I have not seen specified along the manuscript is the sensitivity of these computations on how the authors have defined the different currents studied, i.e., which are the horizontal and vertical boundaries of each current and what criteria were the authors following.

*We have added a paragraph to Section 2.2 to describe the principles of how the currents are defined in ECCOv4-r4 (l.139-145). We have also added a section to the Supplementary detailing the transports section definitions in ECCOv4-r4. We stay as close to the observational transects as the coarse grid allows, and use the same definitions of the currents as in observations (given that it is still meaningful when investigating the ECCOv4-r4 velocity transects).*

- On another note, the instrumentation and methodology used for each monitoring observing system is quite diverse. The authors could consider adding columns for the instrumentation used for each timeseries and the frequency of the observations.

*The details of the measurement systems are too complex to fit into Table 1. The measurement systems are described in Section 2.1 to some detail, while the full description of instrumentation and transport estimation methodologies for all six observing systems can be found in the literature referred to in Section 2.1.*

• Lines 98-99: 'the Oleander record has different temporal resolution than the other time series displayed in Figure 2'. This is the first mention of the time resolution of the time series, so to make this statement the authors should include that information when describing the rest of time series. This is related to the beginning of section 2.3, where it is stated that 'For the observational records with a higher-than-monthly temporal frequency…' and then they specify which ones those are. It can be a bit confusing, so I recommend specifying when describing each dataset.

*To avoid confusion on the resolution of the Oleander measurements we now state that it is estimated in 1-year segments stepped at half-year intervals both when describing the Oleander measurements in Section 2.1 (l.98) and when explaining the data treatment in Section 2.3 (l.170-171).*

• Figure 3. This is up to the authors, but I encourage them to add figure 3 as another column to figure 2, so that it will be referenced to the positions in the map and comparable to the individual currents.

*We see the reviewer's perspective, but we have chosen to keep the layout of Figure 2 and Figure 3 as is. Aligning the panels in one figure according to the observational transects will leave empty spaces between the RAPID panels and the OSNAP panels, which we found not to improve the presentation of figure.*

For the GSR, there is a sharp difference in the behaviour of the overflow between the two intervals: could the authors briefly describe why is that? Are there any difference in instrumentation or methodology on all or any of the sections included in GSR? Why are there two intervals for the overflow (Figure 3) and not for the current (Figure 2)?

*The Greenland-Scotland Ridge inflow branches are measured at different sections than the overflow branches (l.107-109, l.114-116). Both records have several gaps, but the gaps in the GSR inflow record are shorter than 6 consecutive months and have been interpolated over (l.167). The gap in the overflow record during 2006/2007 is more extensive (10 month gap in the Denmark Strait overflow record), which is why the time series is treated as a discontinuous time series both in the analysis and when plotting.*

*As seen in Figure 3, and similarly in Jochumsen et al. Figure 15, the overflow time series may appear different before and after the 2006/2007 gap. One reason is potentially that the Denmark Strait overflow and Faroe Bank Channel overflow were in antiphase until 2002 and in phase afterwards (Jochumsen et al. 2017; Figure 14). Another reason is possibly that the early period has only one mooring measuring the Denmark Strait overflow during 1996-1999 and 2003-2005 as opposed to two moorings in the later period. As we do not perform correlations with the overflow time series and mainly show Figure 3 for putting our findings in context with observations of overturning, we have decided not to expand on these measurement details in the manuscript.*

• Lines 107-115. This paragraph is about the GSR and each component. The description of IFR, FSC and DS can be complemented with a call for Figure S3a to show that the anomalies respond to that of FSC, even if the mean transport is larger over IFR.

*As the reviewer suggests, we have added a sentence on this in l.112-114.*

• Line 162. The authors have normalized the datasets, which can be useful when comparing variables with different units. Could the authors discuss briefly this choice instead of just computing anomalies?

*We agree that this makes for a more intuitive comparison with ECCOv4-r4. Based on this comment and similar comments from Reviewer 1, we have chosen to alter Figure 4 so that panel a) shows the volume transport anomalies from the time mean instead of the normalized volume transport.*

• Lines 218-226. This is a good paragraph discussing reconstructions of AMOC with observations. When discussing inverse models, the recent paper by Fu et al., (2020) could be included, where the authors find no trends in AMOC creating boxes between 24N and 55N.

*The reference to Fu et al. (2020) has been added when discussing AMOC estimates from inverse models (l.248).*

• Table 3. As stated above, even if we don't have trends here for AMOC from ECCO, this is a better place for trends than Table 1. Also, these trends should be expressed with their uncertainties.

*Following the reviewer suggestion, the trends have been moved from Table 1 to Table 3. We have added a figure to the supplementary (Figure S6) showing the 95% confidence interval for the estimated trends in observations and in ECCOv4-r4. Whether trends are significant or not is indicated throughout the manuscript.*

• Lines 317-322: This is a good discussion on the comparison between ECCO and observations. However, it has not been mentioned previously in the manuscript, so the authors may consider placing it above and not in the 'Summary and conclusions' section. One opportunity could be between lines 201 and 202, after discussing that ECCO finds more patterns than observations and before discussing the Ekman layer.

*This point is emphasized in l.216: "In contrast to the observational records, some more distinct patterns of coherence are found within the ECCOv4-r4 estimate (Table 2)". We also come back to this in Section 4 (l.348-353).*

TECHNICAL CORRECTIONS

*We thank the reviewer for also finding the time to help improving the manuscript in terms of text and figure precision. It is most appreciated.*

• Throughout the manuscript, please make sure that the main currents cited and discussed in the text are defined at their first mention and at the appropriate figures and tables.

*We have checked that we have appropriate references when introducing a new current or observational system.*

• Please make sure that the figures in Supplementary Information are in order of appearance in the main text.

*We have changed the order of the supplementary figures based on when they first are mentioned in the text.*

• Figure 1. I find this figure useful to illustrate the introduction, but the authors could include some extra details to make it more accessible.

- On the schematic map in A, only the Gulf Stream is specified with its full name. However, that is not the case for DWBC, EGC, NAC and NwASC. I understand there's not much space in the figure, but it would be useful to define the acronyms at least in the figure caption. NAC, EGC and DWBC are common enough, but that's not the case for NwASC.

  *The full current names with the abbreviations (DWBC, NAC, NwASC, EGC) have been added to the Figure 1 caption.*

- I suggest adding the A16 cruise track from subplot B to the map in A.

  *This is a good suggestion. In trying, we found that it made the map in Figure 1a quite busy. In the end we decided against altering it.*

- On subplot B, it could be helpful to employ the same colours for the arrows representing the upper (purple) and deep (black) circulation.

  *This has also been tested, but due to the reduced color contrast in the upper-ocean we have decided to keep the arrows in black.*

• Line 34. 'of which the Gulf Stream and the extensions' changed to 'of which the Gulf Stream and its extensions'

*This has been changed accordingly.*

• Lines 53-54: the authors could describe shortly the location of the RAPID and OSNAP array (subtropical and subpolar or 24N and 55N). E.g.: The AMOC strength has been measured by cross-basin observing systems at 24N since 2004 (RAPID; Cunningham et al., 2007) and at 55N since 2014 (OSNAP; Lozier et al., 2017).

*As the reviewer suggests, we have reformulated so that the latitudes of RAPID and OSNAP are stated in l.53-54.*

• Line 79. Please specify: 'The mean 32 Sv transported by the Florida Current and the, on average, 4.7 ± 7.5 Sv in the Antilles Current …'

*The ± 7.5 Sv in the Antilles Current in Meinen et al. (2019) is the standard deviation of daily means. We have decided to only include standard deviations of the monthly mean to be consistent throughout the manuscript. We therefore have reformulated the paragraph so that we state the*

*mean transport and standard deviation of the Florida Current and of the total Western Boundary Current transport at RAPID (l.80-82).*

• Figure 2. It could be very helpful to include the name of the observing systems in the map, with the color legend applied for the time series, even if it were just the acronyms and they were defined in the figure caption.

***We have added the section names to the map in Figure 2 as suggested by the reviewers. Legend acronyms are defined in the figure caption.***

• Line 119: 'a single current meter at 100m depth' is missing a space in '100 m'.

***This has been corrected.***

• Line 142: 'ECCOv4-r4 captures the observed peak in $moc\sigma$ in 2015/16 (Figure 3), but the observational time series is too short to get a fair assessment of how well interannual variability is represented at OSNAP.' I understand that this sentence refers to the peak in MOC observations from Figure 3, but it turns out a bit confusing, as there's no ECCO values to compare it against. I would refer readers only to Figure S2c, where the authors can specify the short overlap between both time series (2014-2017).

***We have reformulated the sentence to more precisely express that it is the overlapping time period between ECCOv4-r4 and observations that is too short (l.151-152). Figure S4c (previous Figure 2Sc) is now referred to instead of Figure 3.***

• Lines 151-152: 'compared to observations (0.9 Sv and 3.8 Sv in observations, respectively)'. The second 'in observations' is redundant: 'compared to observations (0.9 Sv and 3.8 Sv, respectively)'.

***This has been corrected.***

• Figure 4. The y-axis labels should include magnitude and unit following the same structure as before: a) STD and b) VT [Sv].

***We have altered the y-axis labels of Figure 4 to be consistent with the other figures.***

• Line 165. Reference to Figure S4 appears before Figure S3.

***We have changed the order of the supplementary figures based on when they are mentioned in the text.***

• Line 236. Reference to Figure S3c instead of only Figure S3.

***This has been changed accordingly (now Figure S2c, l.260).***

• Line 258-259: 'onto the annual mean volume transport time series in ECCOv4-r4 (Figure 5, Figure S7) and in observations (Figure S6)'. I think the references for the supplementary information figures are wrong: 'series in ECCOv4-r4 (Figure 5, Figure S8) and in observations (Figure S7)'.

***The reviewer is right - the figure numbers have now been corrected (l.282-283). The order of the figures is also changed so that it fits with the when the figures are introduced in the text.***

• Line 279: 'indecies' changed to 'indices'.

***This has been corrected.***

References for this review:

Buckley, M. W., & Marshall, J. (2016). Observations, inferences, and mechanisms of the Atlantic Meridional Overturning Circulation: A review. Reviews of Geophysics, 54(1), 5–63. https://doi.org/10.1002/2015RG000493

Fu, Y., Li, F., Karstensen, J., & Wang, C. (2020). A stable Atlantic Meridional Overturning Circulation in a changing North Atlantic Ocean since the 1990s. Science Advances, 6(48), eabc7836. https://doi.org/10.1126/sciadv.abc7836

Han, L. (2023). Exploring the AMOC Connectivity Between the RAPID and OSNAP Lines With a Model-Based Data Set. Geophysical Research Letters, 50(19), 1–10. https://doi.org/10.1029/2023GL105225

Zhao, J., Bower, A., Yang, J., Lin, X., & Holliday, N. P. (2018). Meridional heat transport variability induced by mesoscale processes in the subpolar North Atlantic. Nature Communications, 9. https://doi.org/10.1038/s41467-018-03134-x

Zou, S., Lozier, M. S., & Buckley, M. (2019). How Is Meridional Coherence Maintained in the Lower Limb of the Atlantic Meridional Overturning Circulation? Geophysical Research Letters, 46(1), 244–252. https://doi.org/10.1029/2018GL080958

Zou, S., Lozier, M. S., & Xu, X. (2020). Latitudinal structure of the meridional overturning circulation variability on interannual to decadal time scales in the North Atlantic Ocean. Journal of Climate, 33(9), 3845–3862. https://doi.org/10.1175/jcli-d-19-0215.1